# Chronic High-Dose Neonicotinoid Exposure Decreases Overwinter Survival of *Apis mellifera* L.

**DOI:** 10.3390/insects11010030

**Published:** 2019-12-31

**Authors:** Sarah C. Wood, Ivanna V. Kozii, Igor Medici de Mattos, Roney de Carvalho Macedo Silva, Colby D. Klein, Ihor Dvylyuk, Igor Moshynskyy, Tasha Epp, Elemir Simko

**Affiliations:** 1Department of Veterinary Pathology, Western College of Veterinary Medicine, University of Saskatchewan, 52 Campus Drive, Saskatoon, SK S7N 5B4, Canada; ivanna.kozii@usask.ca (I.V.K.); igor.mattos@usask.ca (I.M.d.M.); roney.macedo@outlook.com (R.d.C.M.S.); colby.klein@usask.ca (C.D.K.); dvylyuk@ukr.net (I.D.); igm800@mail.usask.ca (I.M.); elemir.simko@usask.ca (E.S.); 2Department of Large Animal Clinical Sciences, Western College of Veterinary Medicine, University of Saskatchewan, 52 Campus Drive, Saskatoon, SK S7N 5B4, Canada; tasha.epp@usask.ca

**Keywords:** thiamethoxam, clothianidin, sublethal, honey bee, winter, Canada

## Abstract

Overwinter colony mortality is an ongoing challenge for North American beekeepers. During winter, honey bee colonies rely on stored honey and beebread, which is frequently contaminated with the neonicotinoid insecticides clothianidin and thiamethoxam. To determine whether neonicotinoid exposure affects overwinter survival of *Apis mellifera* L., we chronically exposed overwintering field colonies and winter workers in the laboratory to thiamethoxam or clothianidin at different concentrations and monitored survival and feed consumption. We also investigated the sublethal effects of chronic thiamethoxam exposure on colony pathogen load, queen quality, and colony temperature regulation. Under field conditions, high doses of thiamethoxam significantly increased overwinter mortality compared to controls, with field-realistic doses of thiamethoxam showing no significant effect on colony overwinter survival. Under laboratory conditions, chronic neonicotinoid exposure significantly decreased survival of winter workers relative to negative control at all doses tested. Chronic high-dose thiamethoxam exposure was not shown to impact pathogen load or queen quality, and field-realistic concentrations of thiamethoxam did not affect colony temperature homeostasis. Taken together, these results demonstrate that chronic environmental neonicotinoid exposure significantly decreases survival of winter workers in the laboratory, but only chronic high-dose thiamethoxam significantly decreases overwinter survival of colonies in the field.

## 1. Introduction

Honey bee colony mortality is most prevalent during the winter months in temperate climates, and, since 2007, Canadian beekeepers have experienced average overwinter losses in excess of the 15% economically sustainable threshold [1]. In winter 2019, Canadian beekeepers experienced 25.7% overwinter colony loss on average, with beekeepers attributing their losses to weather, poor queen quality, weak fall colonies, *Varroa* infestation, and starvation [1].

Pesticide exposure through stored honey and pollen is another potential stressor contributing to overwinter colony loss. Canola, also known as oilseed rape, is one of the most common bee-attractive crops grown in Canada, and most of this canola is grown from neonicotinoid-treated seed [2]. The neonicotinoid insecticides clothianidin (CLO) and thiamethoxam (THI) are commonly detected in pollen, nectar, and honey, at mean concentrations from 1.9–9.4 ng/g of CLO and 6.4–28.9 ng/g of THI [3]. In Saskatchewan, Canada, CLO was detected in 68% of honey samples at mean doses of 8.2 ng/g, and THI was detected in 75% of honey samples at a mean of 17.2 ng/g [4].

Previous chronic colony-feeding studies [5], laboratory studies [6,7], and field trials [8] have examined the effects of THI or CLO exposure on overwinter survival of honey bees. A chronic, summer colony-feeding study [5] found significant decreases in overwinter survival with 100 ng/g THI exposure, and no significant effect of 12.5–50 ng/g THI exposure on overwintering. Interestingly, winter workers were more sensitive to chronic neonicotinoid exposure under laboratory conditions, with doses of 20 and 50 ng/g of CLO significantly reducing survival of winter adult workers and no effect of 1–10 ng/g of CLO on laboratory survival [6]. Similar to the laboratory results, field trials demonstrated that colonies exposed to <1–7 ng/g of THI [8] or 0.5–2 ng/g of CLO [9] during foraging show no significant difference in overwinter survival relative to controls [8,9]. Although a link between field-realistic THI and CLO exposure and overwinter colony mortality has not been established thus far, the effects of neonicotinoids on overwintering honey bees warrant further study.

Winter worker honey bees have important endocrine and metabolic differences from summer adult workers, which increase their lifespan by 6–8 times, but may also alter their susceptibility to pesticides. For example, compared to summer foragers, winter workers have low levels of juvenile hormone and high levels of vitellogenin and other proteins in the hemolymph [10,11,12,13,14]; decreased protein synthesis, transport across the midgut, and catabolism [15,16,17]; and lower activity of monooxygenase enzymes important for pesticide detoxification [18]. It remains unclear how the physiologic differences between summer and winter workers alter pesticide sensitivity, with one study demonstrating increased susceptibility of winter workers to acute THI and CLO exposure compared to summer workers [7], and another study showing decreased sensitivity of winter workers to acute synergistic effects of a pyrethroid insecticide and an imidazole fungicide [19].

Few studies have examined the correlation between field and laboratory effects [20] of chronic neonicotinoid exposure on honey bee workers, particularly in Saskatchewan, Canada, where outdoor colonies must endure severe cold during the winter months, and during summer, colonies commonly forage on THI- or CLO-treated canola. We hypothesize that chronic dietary neonicotinoid exposure will decrease overwinter survival of (1) honey bee colonies in the field and (2) adult winter honey bee workers in the laboratory during the Saskatchewan winter. In the field, we hypothesize that chronic THI exposure will decrease colony size, queen quality, and colony temperature regulation.

We performed both field and laboratory experiments, to examine the effects of chronic overwinter neonicotinoid exposure on overwinter survival at field-realistic and high-dose neonicotinoid concentrations during the harsh climactic conditions of Saskatchewan winters. In the field experiments, we also investigated the sublethal effects of chronic THI exposure on colony pathogen load, queen quality, and colony temperature regulation.

## 2. Materials and Methods

### 2.1. Experimental Design

Two field trials (F1 and F2) and two laboratory cage trials (C1 and C2) were conducted (Figure 1). The overall objective was to examine the effect of neonicotinoid exposure on colony overwintering. F1 was designed to test 100 ng/g of THI as a positive control, as well as 20 ng/g of THI, which is a high, environmentally realistic dose for Saskatchewan [4]. C1 was designed as a laboratory model of F1 to test the same concentrations of THI and its metabolite CLO, with the addition of a 5 ng/g dose group. Based on the overwinter survival results of F1, we designed F2 to examine whether weak fall colony strength, in combination with field-realistic 5 and 10 ng/g THI exposure, would predispose colonies to overwinter mortality. We designed C2 as a laboratory model of both F1 and F2 by including all of the doses tested in the field for both THI and its metabolite CLO. To understand sublethal effects of neonicotinoid exposure on overwintering, we evaluated pathogen levels and queen quality in F1. Based on the absence of sublethal effects observed in F1, we chose a different sublethal outcome (temperature) for F2. Thus, the field and laboratory trials should not be considered replicates, and each trial had an appropriate negative control group for comparison.

### 2.2. Field Trials

To compare the effects of THI exposure on overwinter colony survival, field colonies received ad libitum control or THI-contaminated 2:1 (w:v) sucrose syrup over 5 weeks in fall, and overwinter survival was assessed the following spring (Table 1). F1 consisted of sixty strong colonies, each equalized to two brood chambers and randomized into treatment groups, with newly mated sister queens from one genetic lineage. F2 contained sixty-four weak colonies with newly mated sister queens from two genetic lineages which were unrelated to the F1 queens. The F2 colonies occupied 1 to 2 brood chambers and were stratified across treatment groups. The F1 colonies were not reused in F2. Both F1 and F2 occurred on land rented or owned by the University of Saskatchewan Goodale Research and Teaching Farm (Clavet, Sk, Canada), at two different yards, which were 2.3 km apart (52°01′50.6′′ N 106°32′26.6′′ W in F1 and 52°02′34′′ N 106°30′45′′ W in F2).

To prepare the THI-treated syrup, a stock of THI in water was prepared at a concentration of 100 (F1) or 10 µg/mL (F2), and the appropriate volume of stock was added to 2:1 sucrose solution, to achieve the desired concentration in ng THI per g of syrup. Stock was prepared using analytical standard THI (product number 37924; batch number BCBT3749; purity 99.7%; expiry November 2021; obtained from MilliporeSigma Canada Co., Oakville, ON, Canada). Syrup was mixed thoroughly prior to feeding. In F1, colonies were administered sugar syrup through 4 L top-fed jars initially, followed by 4 L frame feeders. In F2, colonies received syrup exclusively through frame feeders. Any syrup remaining in the frame feeder in the spring was accounted for when calculating total syrup consumption. In early October, the colonies were administered one final feeding of experimental syrup, and the colonies were wrapped in groups of four for winter, using standard side wraps (R4 thermal rating) and insulating top pillows (R8 thermal rating) with a plywood top cover. Preliminary overwinter colony survival was assessed in April (F1) or March (F2) and final overwinter survival was determined at unwrapping in May.

All study colonies were weighed at three time points: (1) prior to syrup-feeding, (2) prior to winter wrapping, and (3) after spring unwrapping, using a mechanical hanging scale (Salter Model 235, Brecknell Scales, Fairmont, MN, USA) with an accuracy to the nearest 0.5 kg.

To estimate population size, the study colonies were ‘cluster sized’ at four time points: (1) prior to syrup-feeding (F2 only), (2) prior to winter wrapping, (3) during the initial spring survival assessment, and (4) after spring unwrapping. Briefly, using a 16.2 megapixel Nikon D7000 digital camera (Minato, Tokyo, Japan) with a Nikon 18 ± 105 mm lens, a photo was taken of the adult bee cluster on the tops of each occupied super early in the morning prior to bees flying out to forage. The cluster size for each super was estimated by counting the number of interframe spaces to the nearest 0.25 occupied by adult bees in the photos. The overall cluster size for each colony was obtained by summing the cluster size for each super in the colony [21]. At time point 3 in F1 and F2, the cluster size was based on a photo of the top super only because the second super could not be accessed due to the winter wrap.

#### 2.2.1. Pathogen Monitoring in F1

In F1, all colonies were sampled for phoretic *Varroa* mites before (August 2017) and after (October 2017) treatment with Amitraz, as well as at the beginning and the end of May 2018, using an alcohol wash [22] of approximately 300 workers from a brood frame. Briefly, workers were sampled in 200 mL of windshield washer fluid (Turbo Power^®^, All Season Windshield Washer; Recochem Inc., Edmonton, AB, Canada) or methyl hydrate (Turbo Power^®^ Heavy Duty 99.9% Pure; Recochem Inc., Edmonton, AB, Canada) and shaken for 30 min on a rotary shaker, at 200 rpm [23]. Bees were strained from the wash fluid, and the mites were counted to obtain a percent infestation (mites/bees sampled × 100%). In June 2018, a 17 × 6 cm area of capped brood was uncapped, to examine for the presence of *Varroa* in a subset of the surviving colonies (11 colonies from control group; 10 colonies from 20 ng/g group, and 7 colonies from 100 ng/g group).

Additionally, all F1 colonies were sampled for *Nosema* spore counts in September 2017 and May 2018. *Nosema* spore counts per bee were generated by macerating 60 workers sampled from a honey frame (fall 2017) or the entrance (spring 2018) of each colony in 60 mL of phosphate-buffered saline for 1 min, using a Stomacher^®^ 80 Biomaster (Seward, Davie, FL, USA), and counting spores in 0.02 mm^3^, using a haemocytometer (Hausser Scientific, Horsham, PA, USA) and a phase-contrast microscope (Olympus IX51, Tokyo, Japan) [24]. Two samples of macerate were counted per colony, and the results averaged to obtain a spores per bee count for each colony.

#### 2.2.2. Queen Quality in F1

Quality of the queens in F1 was assessed by sacrificing and weighing six queens from each control and treatment group in August 2018. Each queen’s spermatheca was diluted in 1 mL of Kiev buffer (sodium citrate dihydrate 24.3 g/L, NaHCO_3_ 2.1 g/L, KCl 0.4 g/L, and sulphanilamide 0.3 g/L, D-(+) glucose 3.0 g/L in double-distilled water, all from MilliporeSigma Canada Co., Oakville, ON, Canada) [25], and total spermatozoa counts were performed by using a 1:16 dilution of spermatozoa with a hemocytometer (Hausser Scientific, Horsham, PA, USA) and a light microscope (Olympus CX22, Tokyo, Japan) [24]. Sperm viability was assessed by staining 50 µL of spermatozoa in Kiev buffer with SYBR^®^14 and propidium iodide (LIVE/DEAD^TM^ Sperm Viability Kit, ThermoFisher Scientific, Waltham, MA, USA) and counting live and dead sperm in a minimum of 10 and maximum of 20, 20× fields to reach a minimum count of 200 sperm per sample, using a fluorescent compound microscope (Olympus BX51, Tokyo, Japan) [26].

#### 2.2.3. Temperature Monitoring in F2

In F2, a Thermochron iButton (DS1921G-F5#, Embedded Data Systems, Lawrenceburg, KY, USA), vacuum-packed in a plastic strip, was inserted through the top entrance of each colony on 19 February 2019, to monitor hourly, within-colony temperature, until 8 May 2019 (78 days).

#### 2.2.4. Analysis of Data

All statistical analyses were performed, using Stata/SE 15.1 (College Station, TX, USA) with *p* < 0.05 considered significant. Data are presented as medians or means ± standard deviation (SD). Overwinter survival was analyzed by Chi-square and a z-test. Syrup consumption, colony weight gain, queen weight, sperm viability, and sperm counts were analyzed by using a one-way ANOVA with a Bonferroni multiple comparison test. Cluster size at each time point was analyzed using a one-way ANOVA with a Bonferroni multiple comparison test or a Kruskal-Wallis equality of populations rank test with a Dunn’s pairwise comparison test. *Nosema* and *Varroa* data were analyzed using a Kruskal-Wallis equality of populations rank test.

Temperature data were analyzed from surviving colonies in spring 2019 which retained the sensor within the colony throughout the temperature monitoring period (14 colonies in control group, 10 colonies in 10 ng/g group and 11 colonies in 5 ng/g group), excluding days where colonies were opened for cluster sizing or spring treatments. In accordance with Meikle et al. (2016) [27], a running average temperature was calculated for the 12 h before and after each temperature measurement, as well as a detrended temperature, calculated by subtracting the running average from each temperature measurement. For each day, the minimum and maximum running average and detrended temperatures were determined for each colony and compared across treatment groups, using a linear mixed model. The assumptions of the model were met.

### 2.3. Laboratory Cage Trials

To evaluate the effects of neonicotinoids on laboratory survival of winter adult workers, winter workers received ad libitum 1:1 (w:v) sucrose syrup containing THI or CLO, over 30 days, and survival was monitored daily (Table 2). Negative controls received untreated syrup, and positive controls received syrup containing dimethoate [28].

Adult workers were sampled from a single outdoor, overwintering, queenright colony that was derived from three colonies of different genetic lineages, which were merged one week prior to the beginning of C1. The colony was placed indoors at 15 degrees, 12 h prior to sampling. The colony did not have brood at the time of sampling for C1 or C2, likely due to unusually cold spring weather. In late March, prior to sampling for C2, the colony was treated for *Varroa* mites with Amitraz-impregnated strips (Apivar^®^, Veto-pharma, Palaiseau, France), in accordance with label instructions.

In each trial, stainless steel insect cages (measuring 7.5 × 4 × 5.5 cm; Small Life Supplies, Cambridgeshire, Great Britain) were filled with adult worker honey bees, each, by gently vacuuming bees directly from the frame into a cage. After collection, prior to the start of each trial, the bees were given a 24-h acclimatization period, in which they received untreated 1:1 (*w/v*) sucrose solution. The cages were kept in darkness, within an incubator, at 29 °C and 60% relative humidity. In each trial, the cages were randomly assigned to treatment and control groups (Table 2). Each day, the number of dead bees in each cage was recorded, and the dead bees were removed. All procedures were performed under red light.

To ensure equal numbers of active molecules of THI and CLO at each concentration tested, concentrations (100, 20, 10, and 5 ng/g) were converted to nanomolar (400, 80, 40, and 20 nM). Thus, the actual ng/g doses of THI and CLO tested are presented in Table 2; however, for simplicity we will refer to the test doses as 100, 20, 10, and 5 ng/g in the text and figures. A 10 µg/mL stock of THI or CLO in water was prepared, and the appropriate volume of stock added to sucrose solution to achieve the desired concentration in nM. Similarly, for the positive control, a 100 µg/mL stock of dimethoate in water was prepared and diluted in sucrose solution to a concentration of 1000 ng/g. Stocks were prepared by using analytical standard pesticides from MilliporeSigma Canada Co., Oakville, ON, Canada. For each pesticide, the name, product number, batch number, purity, and expiration date are listed: (i) thiamethoxam, 37924, BCBT3749, 99.7%, November 2021 (ii) clothianidin, 33589, BCBS3968V, 99.9%, June 2020 (iii) dimethoate, 45449, BCBS9338V, 99.8%, August 2021.

Fresh treatment solution was provided every third day, and the feeding syringes were weighed pre-insertion and post-removal, to monitor diet consumption. Three cages without bees were used to monitor diet evaporation in each trial.

### 2.4. Analysis of Data

All statistical analyses were performed by using Stata/SE 15.1 (College Station, TX, USA), with *p* < 0.05 considered significant. Data are presented as means ± standard deviation (SD). For each trial, syrup consumption (mean grams per bee per 3 days) corrected for evaporation was analyzed using a generalized estimating equation (GEE) population averaged model with an exchangeable correlation structure and time and treatment as independent variables. Syrup consumption of the positive control was not included in the analysis.

Survival over 30 days for each trial was modeled by using a Weibull hazard function, with an accelerated failure time model. The survival data were clustered by cage, with bee survival considered non-independent between bees in the same cage and bee survival considered independent between bees in different cages. Cox-Snell residuals, Martingale residuals, deviance residuals, and proportionality of hazards were assessed graphically, to evaluate the goodness-of-fit of the model, the functional form of the model, the presence of outliers, and the model assumptions, respectively.

## 3. Results

### 3.1. Effects of Thiamethoxam on Overwintering Colonies of Apis mellifera in Field Trial 1 (F1)

Strong fall colonies exposed to high dose, field unrealistic concentrations of 100 ng/g of THI experienced significant (z = 3.6, *p* < 0.001), 55% greater overwinter mortality relative to control colonies (Figure 2a) and significant (F_2,37_ = 4.59, *p* = 0.045), two interframe space decreases in early spring adult bee cluster compared to the control (Figure 3a). A dose response was observed for overwinter colony survival (Figure 2a), with 10% overwinter loss of control colonies (2/20), 25% overwinter loss of medium dose colonies (5/20), and 65% overwinter loss of high dose colonies (13/20). Strong fall colonies chronically exposed to high environmental doses of 20 ng/g of THI overwinter did not experience significant increases in overwinter mortality relative to the control (z = 1.2, *p* = 0.2119).

During fall feeding, the twenty 100 ng/g THI-treated colonies consumed significantly less syrup (14.25 L, SD = 4.11 L, Figure 4a) compared to the twenty control (24.97 L, SD = 4.04 L, F_2,57_ = 37.65, *p* < 0.001) and twenty 20 ng/g THI-treated colonies (21.89 L, SD = 3.92 L, *p* < 0.001). Furthermore, the twenty 100 ng/g THI-treated colonies lost significantly less weight (15.85 kg, SD = 4.65 kg) from October 2017 to May 2018 compared to the twenty control colonies (19.3 kg, SD = 4.19 kg, F_2,57_ = 3.34, *p* = 0.044, Figure 5a).

Prior to THI exposure, all colonies had low levels of *Varroa* (0.36% infestation, SD = 0.38) and *Nosema* (5.21 × 10^4^ spores/bee, SD = 1.55 × 10^5^). In October 2017, mean percent *Varroa* infestation declined to 0.02% (SD = 0.082) after Amitraz treatment, with no significant difference in infestation across treatment groups (X^2^(2) = 0.295, *p* = 0.8628). In early spring 2018, there were no significant THI-treatment effects for *Nosema* infection (6.4 × 10^5^ spores/bee, SD = 1.42 × 10^6^; X^2^(2) = 4.098, *p* = 0.1289) or *Varroa* infestation (phoretic varroa not detected in any colony). In late spring 2018, after Amitraz treatment, mean percent *Varroa* infestation was 0.014% (SD = 0.063), with no treatment effect on infestation (X^2^(2) = 0.358, *p* = 0.8362). *Varroa* was not observed in any of the capped brood examined in the control and THI-treated colonies.

Queen quality was not significantly affected by chronic overwinter THI exposure in F1 (Figure 6). Six control queens, six queens exposed to 20 ng/g of THI, and six queens exposed to 100 ng/g of THI did not differ significantly in sperm viability (mean live = 69.4%, SD = 13.69, F_2,15_ = 3.23, *p* = 0.0681, Figure 6a), total sperm count (2.91 × 10^6^, SD = 1.35 × 10^6^, F_2,15_ = 1.98, *p* = 0.1729, Figure 6b), or queen weight (242 mg, SD = 26.62, F_2,15_ = 3.15, *p* = 0.0722, Figure 6c).

### 3.2. Effects of Thiamethoxam on Overwintering Colonies of Apis mellifera in Field Trial 2 (F2)

Weak fall colonies chronically exposed to environmental (5 or 10 ng/g) doses of THI did not experience significant decreases in overwinter colony survival (X^2^(2) = 0.743, *p* = 0.699; Figure 2) or colony cluster size relative to control (Figure 3). Prior to THI exposure, the colonies in F2 were weaker than the colonies in F1. The mean cluster size in September 2018 in F2 was 8.37 interframe spaces (SD = 2.69, Figure 3b), while the mean frames of bees in September 2017 for F1 was 13.04 frames (SD = 3.10). Not surprisingly, the overwinter mortality of controls in F2 (36%, Figure 2b) was over four times greater than the overwinter mortality of 112 non-study colonies in our research apiary in winter 2018–2019 (8.9%).

During fall feeding, there was no significant difference (F_2,61_ = 1.42, *p* = 0.2501) in syrup consumption (18.6 L, SD = 3.50 L) of the 64 control and treatment colonies (Figure 4b). Additionally, there was no significant difference (F_2,61_ = 0.58, *p* = 0.5616) in overwinter weight loss from October 2018 to May 2019 (14.27 kg, SD = 3.96 kg) of the 64 control and treatment colonies (Figure 5b).

The in-hive temperature for colonies exposed to 5 or 10 ng/g of THI was not significantly different from controls (Figure 7). The maximum and minimum running average daily temperatures (29.06 °C, SD = 4.49 and 13.44 °C, SD = 10.14, respectively; Figure 7d) did not differ significantly across treatment groups (X^2^(2) = 0.27, *p* = 0.873 for maximums and X^2^(2) = 0.53, *p* = 0.7677 for minimums). Similarly, there was no significant effect of THI treatment on the maximum and minimum detrended daily temperature amplitudes (3.54 °C, SD = 2.12 and −4.47 °C, SD = 3.23, respectively; Figure 7b; X^2^(2) = 3.06, *p* = 0.2163 for maximums and X^2^(2) = 1.36, *p* = 0.5077 for minimums).

The in-hive maximum and minimum running average and detrended temperature amplitudes varied significantly over time during F2 (*p* < 0.001; Figure 7), with no significant interaction between THI treatment and time (X^2^(154) = 89.14, P = 1.0 for maximum and X^2^(154) = 77.35, *p* = 1.0 for minimum running average temperature; X^2^(154) = 82.87, *p* = 1.0 for maximum and X^2^(154) = 169.38, *p* = 0.1877 for minimum detrended temperature amplitude).

### 3.3. Effects of Chronic Thiamethoxam or Clothianidin Exposure of Winter Apis mellifera Adult Workers during Laboratory Cage Trial 1 (C1)

We found that chronic laboratory neonicotinoid exposure significantly (X^2^(6) = 124.73, *p* < 0.001) decreased survival time of winter workers relative to control in a dose-dependent manner (Figure 8a,c and Appendix A). The negative control had a median survival of 16.48 days (Appendix A). As a positive control, 30 workers were treated with 1000 ng/g of dimethoate, resulting in a median survival of 2 days.

For the same dose, there was no significant difference (*p* > 0.05) in survival between THI- or CLO-treated workers (Figure 8a,c). Workers treated with 100 ng/g of neonicotinoids survived a 67%–77% shorter time compared to negative controls; workers treated with 20 ng/g neonicotinoids survived a 38% shorter time relative to negative controls; and workers exposed to 5 ng/g neonicotinoids survived a 17%–20% shorter time compared to negative controls (Figure 8a,c and Appendix A).

Mean syrup consumption was 0.30 g per bee per three days (SD = 0.19 g); thus, consumption was calculated to be 83 µL per bee per day (Figure 9). There was no significant difference in syrup consumption of treatment groups relative to control (X^2^(6) = 11.64, *p* = 0.0706, Figure 9a). Syrup consumption varied significantly over time (X^2^(8) = 26.88, *p* < 0.001), but there was no interaction between time and treatment (X^2^(27) = 29.85, *p* = 0.321).

### 3.4. Effects of Chronic Thiamethoxam or Clothianidin Exposure of Winter Apis mellifera Adult Workers during Laboratory Cage Trial 2 (C2)

We found that chronic laboratory neonicotinoid exposure significantly (X^2^(8) = 167.57, *p* < 0.001) decreased survival time of winter workers relative to control in a dose-dependent manner (Figure 8b,d and Appendix A). The negative control had a median survival of 23.89 days (Appendix A). As a positive control, 30 workers were treated with 1000 ng/g dimethoate resulting in a median survival of 3 days.

For the same dose, from 10–100 ng/g, there was no significant difference (*p* > 0.05) in survival between THI or CLO treated workers. Workers treated with 100 ng/g neonicotinoids survived a 77% shorter time compared to negative controls; workers treated with 20 ng/g neonicotinoids survived a 33%–34% shorter time relative to negative controls; and workers exposed to 10 ng/g neonicotinoids survived a 17%–21% shorter time compared to negative controls (Figure 8b,d and Appendix A). Workers exposed to 5 ng/g of THI survived 15% shorter time than workers exposed to 5 ng/g of CLO (*p* = 0.003) and a 27% shorter time than negative controls (*p* < 0.001). Workers exposed to 5 ng/g of CLO survived a 13.9% shorter time than negative controls (*p* = 0.003).

Mean syrup consumption was 0.23 g per bee per three days (SD = 0.16 g), and, thus, consumption was calculated to be 64 µL per bee per day (Figure 9). There was a significant interaction between time and treatment for syrup consumption (X^2^(51) = 94.55, *p* = 0.0002, Figure 9b), indicating that syrup consumption was different over time, depending on neonicotinoid treatment. Since consumption per bee per three days was calculated by using the final number of living workers in a cage [29], cages which experienced high mortality over the preceding three days (for example, THI 100 in Figure 9b,d) had an elevated consumption per bee value, contributing to the interaction between treatment and time.

## 4. Discussion

In this study, we demonstrated that chronic experimental neonicotinoid exposure during the Saskatchewan winter significantly decreased overwinter survival of (1) honey bee colonies in the field at high doses and (2) adult winter worker honey bees in the laboratory at field-realistic and high doses. Our study shows that colonies overwintering in Saskatchewan on canola honey and beebread are at low risk of mortality from chronic neonicotinoid exposure.

### 4.1. Effects of Thiamethoxam on Overwintering Colonies of Apis mellifera in Field Trial 1 (F1)

At the colony level, a dose response in overwinter survival was observed for colonies chronically exposed to THI overwinter, with no observed effect of chronic overwinter THI exposure on pathogen load or queen quality.

Colony overwinter survival and cluster size was significantly decreased by exposure to 100 ng/g of THI (Figure 2a and Figure 3a). Our results agree with the colony-level feeding studies of Thompson et al. (2019) [5] and Overmeyer et al. (2018) [20], who found that colonies fed 100 ng/g of THI during six weeks in summer had significant (50%) reductions in number of adult bees relative to controls prior to overwintering and a significant, two times increase in overwinter mortality, but colonies fed lower doses of THI (12.5–50 ng/g THI) had no long-lasting colony effects.

Similar to our study, Overmeyer et al. (2018) [20] observed that high dose, 100 ng/g THI-exposed colonies consumed less syrup compared to lower-dose THI treatments and controls. Decreased colony strength and population size may explain the decrease in syrup consumption and overwinter weight loss (reflecting consumption of overwinter food stores) of the 100 ng/g THI-treated colonies in our study (Figure 4a and Figure 5a).

As is typical of most colony-level studies, sample size was the greatest weakness of our field trial. While the number of colonies per treatment in F1 (twenty colonies) was larger than some overwinter studies [5], we still had inadequate statistical power to detect an effect at 20 ng/g of THI. A strength of our study design is that neonicotinoid exposure in our study occurred immediately prior to overwintering rather than during summer, as in other studies [5,20].

Despite an apparent dose response in colony survival (Figure 2a), pathogen load and queen quality were not significantly impacted by overwinter exposure to THI at high (20–100 ng/g) doses (Figure 6). Considering the overall low levels of *Varroa* and *Nosema* in our study colonies throughout F1, it is not surprising that a treatment effect of THI exposure was not observed; however, we cannot rule out synergy or additive effects of neonicotinoids in colonies with higher disease pressure.

The absence of a treatment effect on queen quality in our study is in contrast to other studies [26,30] which have shown that queens are negatively impacted by chronic, colony-level, field-realistic 4–5 ng/g THI and 1–2 ng/g CLO exposure, demonstrating 60% increases in queen supersedure [30], and significant 20% and 9% decreases in total number and viability of spermatozoa, respectively, in queen spermathecae [26]. The discrepancy in queen quality results of our study and others may be explained by differences in timing of colony neonicotinoid exposure. In our study, THI exposure took place in the fall, when the queen is much less reproductively active, in contrast to other studies in which queens were exposed during development [26] or during summer colony build-up [30]. Considering that queen quality is an oft-cited reason for reduced overwintering success [1], our data would suggest that chronic overwinter neonicotinoid exposure is not responsible for declines in queen reproductive health. However, our low sample size (six queens) may have been inadequate to detect treatment effects on queen quality.

### 4.2. Effects of Thiamethoxam on Overwintering Colonies of Apis mellifera in Field Trial 2 (F2)

The combination of weak colony strength in fall and chronic, environmentally realistic, 5 or 10 ng/g THI exposure did not significantly increase overwinter mortality or affect temperature homeostasis relative to control colonies (Figure 2b and Figure 7), suggesting that colonies are resilient in the face of combined stressors. Similar to our findings, Sandrock et al. (2014) [30] showed that colony overwintering success was not affected by chronic 5.3 ng/g THI and 2.05 ng/g CLO exposure. Furthermore, our findings in F2 support the results of a four-year colony monitoring study, which found no correlation between overwinter colony mortality and environmental pesticide residues in bee bread, although the beebread did not contain THI or CLO residues [31]. As in F1 above, despite a sample size per group of twenty-one to twenty-two colonies, we lacked adequate statistical power, to detect a treatment effect on survival in F2.

Sublethal effects of field-realistic THI exposure on colony temperature homeostasis were not observed. Laboratory studies have shown that individual worker bees exposed to sublethal doses of CLO have decreased ability to detect and respond to environmental stimuli, suggesting a potential mechanism for decreased temperature control within an overwintering colony chronically exposed to neonicotinoids [32]. However, our F2 temperature results suggest that effects of neonicotinoids on individual bees may not scale up to cause colony-level dysfunction. The lack of a treatment effect on temperature regulation in our study contrasts with Colin et al. (2019) [33], who demonstrated that colonies chronically exposed to 5 ng/g of the neonicotinoid imidacloprid through sugar syrup had higher average overwinter in-hive temperatures and decreased colony temperature variability, and Meikle et al. (2017) [34] who observed that a history of commercial pollination activity and agrochemical exposure was associated with lower overwinter internal colony temperatures and increased colony temperature variability. These inconsistent results regarding the effect of pesticide exposure on colony temperature regulation may be explained by variation in overwintering climate and beekeeping practices. Our study colonies wintered outdoors in Saskatchewan, experiencing ambient temperatures of −20 °C and below, while Meikle et al. (2017) [34] wintered their study colonies indoors at 7 °C, and Colin et al. (2019) [33] wintered their study colonies outdoors in the desert environment of Tuscon, Arizona. Notably, Colin et al. (2019) [33] were unable to repeat the results of their Tuscon study in an identical trial conducted in Sydney, Australia, underscoring the geographic variation in colony temperature control in response to pesticide exposure.

Considering the absence of lethal or sublethal effects of overwinter, environmental THI exposure demonstrated in our study, parameters other than pesticide exposure should be examined to predict colony overwinter success, including levels of *Varroa*, fall colony strength, deformed wing virus and acute bee paralysis virus titers, queen age, and beekeeper knowledge and experience [31,35].

### 4.3. Effects of Chronic Thiamethoxam or Clothianidin Exposure of Winter Apis mellifera Adult Workers in Laboratory Cage Trial 1 (C1)

In accordance with F1, exposure to high dose, 100 ng/g of THI or CLO in the laboratory resulted in the greatest decrease in median survival time relative to negative controls (*p* < 0.001, Figure 8a,c and Appendix A). However, in contrast to F1, we found significant effects of chronic THI or CLO exposure on adult winter worker laboratory survival at field realistic (5 ng/g) and high environmental (20 ng/g) doses. Thus, our study demonstrates that winter workers in the laboratory are more sensitive to chronic neonicotinoid exposure compared to colonies overwintering in the field. One explanation for this observation could be that the eusocial structure of a colony buffers pesticide stress, while individual workers in the laboratory are rendered more vulnerable to neonicotinoid toxicity due to a lack of eusocial support and the stress of the artificial cage environment [29,36]. Alternatively, in the field, there was likely dilution of the THI administered during fall feeding due to colony consumption of existing brood honey stores overwinter, resulting in decreased THI exposure of workers in the field colonies compared to workers in the laboratory.

### 4.4. Effects of Chronic Thiamethoxam or Clothiandin Exposure of Winter Apis mellifera Adult Workers in Laboratory Cage Trial 2 (C2)

In contrast to F2, significant (*p* < 0.05) decreases in winter worker survival were observed after chronic exposure to 5 and 10 ng/g of THI or CLO in the laboratory (Appendix A). While the combined stress of weak colony strength and neonicotinoid exposure did not predispose THI-treated colonies to overwinter loss in the field, the addition of cage-associated stress may have predisposed winter workers to neonicotinoid-related mortality in the laboratory. For example, the cage-volume-to-bee ratio used in our laboratory trials (17:1) was higher than the 3:1 ratio recommended by others [29], which may have increased the social stress for the workers in our study.

Similar to our findings, Baines et al. (2017) [7] demonstrated significant negative effects of environmental concentrations of THI or CLO on winter adult worker survival in the laboratory. In contrast to our study, Alkassab and Kirchner (2016) [6] found that chronic exposure to 10 ng/g of CLO did not significantly decrease winter worker survival in the laboratory, while we found that winter workers exposed to 10 ng/g of CLO had a significant (21%) decrease in survival time (*p* < 0.001, Appendix A) compared to negative controls. A higher daily syrup consumption in our study (77 mg/bee/day vs. 60 mg/bee/day) and a longer exposure time in our study (30 days compared to 12 days) could explain the differences in survival in our study [6].

Compared to the literature on chronic laboratory neonicotinoid exposure of summer workers, our study demonstrates that winter workers are more sensitive to THI under conditions of chronic exposure in the laboratory. In our study, significant decreases in winter worker survival were observed after chronic exposure to 5, 10, and 20 ng/g of THI, while, in our previous work [37], we found no significant effect of chronic 10 and 20 ng/g of THI exposure on summer worker survival. Similarly, Overmyer et al. (2018) [20] found no effect of chronic, 117 ng/g of THI exposure on summer adult worker survival, while, in our study, chronic, 100 ng/g of THI exposure resulted in over 70% decreases in survival time of winter workers compared to controls (Appendix A).

## 5. Conclusions

In summary, chronic, high-dose, environmentally unrealistic exposure to 100 ng/g of THI was necessary before a significant decrease in overwinter survival of strong fall colonies was observed. This study demonstrated no effect of environmental doses of THI on overwinter survival of weak fall colonies. Considering that the same environmental doses of THI resulted in significant overwinter mortality of winter worker bees in the laboratory, this study highlights the importance of field studies to validate laboratory data.

## Figures and Tables

**Figure 1 insects-11-00030-f001:**
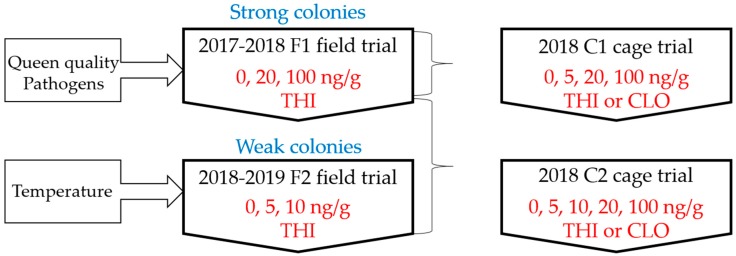
Experimental design of field and laboratory cage trials. Field trials examined the effects of chronic overwinter exposure to thiamethoxam (THI) on survival of strong colonies (F1) or weak colonies (F2), as well as sublethal effects on queen quality and pathogen load (F1) and colony temperature regulation (F2). Cage trials examined the effects of chronic exposure to THI or its metabolite clothianidin (CLO) on laboratory survival of winter adult workers.

**Figure 2 insects-11-00030-f002:**
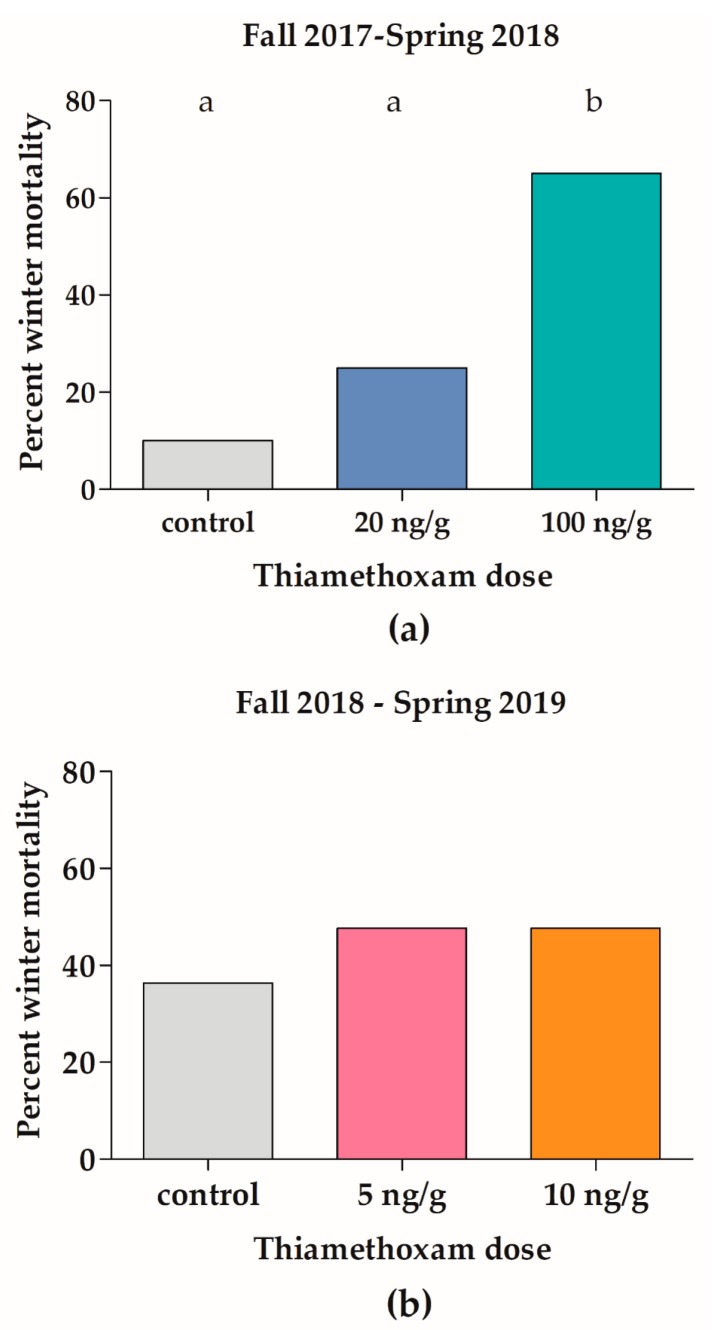
Chronic high-dose (100 ng/g) thiamethoxam significantly increases overwinter mortality of *Apis mellifera* colonies in the field. Honey bee colonies were chronically exposed to thiamethoxam over winter in field trial 1 (2017–2018) (**a**) or field trial 2 (2018–2019) (**b**), and survival of the colonies was evaluated the following spring. Bars indicate percent overwinter colony loss for twenty to twenty-two colonies per group. Different letters indicate significant differences by a z-test, *p* < 0.05.

**Figure 3 insects-11-00030-f003:**
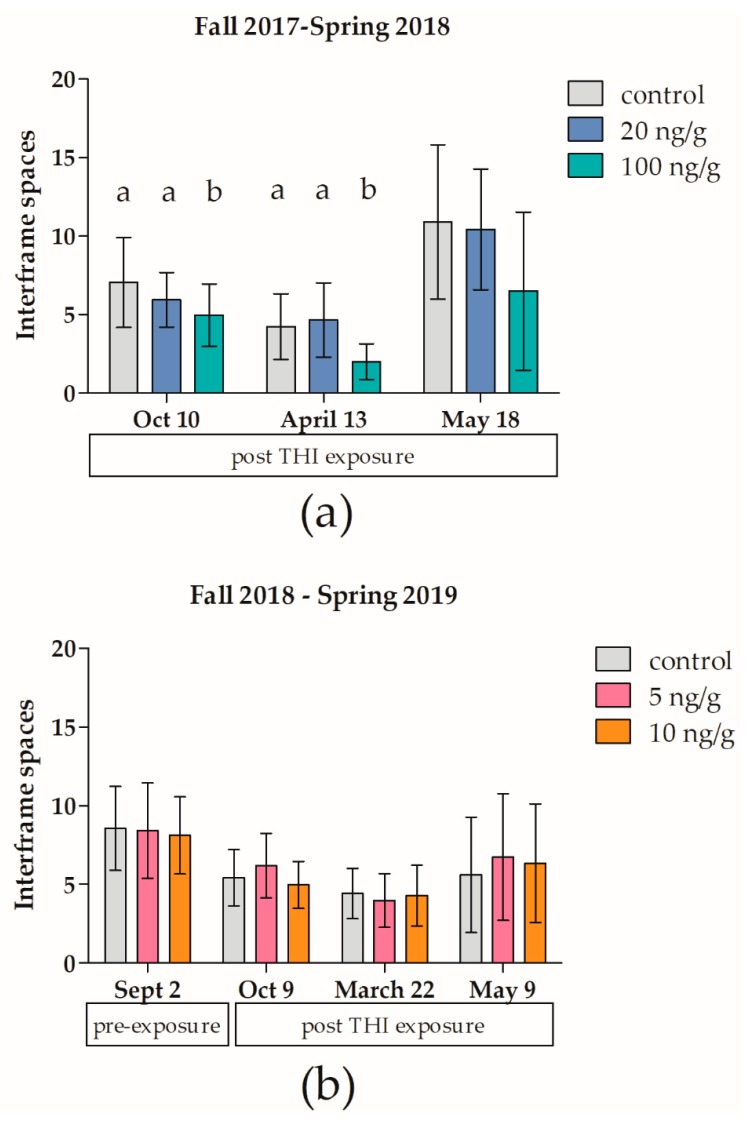
Chronic high-dose (100 ng/g) thiamethoxam significantly decreases cluster size of *Apis mellifera* colonies in the field. Honey bee colonies were exposed to thiamethoxam (THI) over winter in field trial 1 (2017–2018) (**a**) or field trial 2 (2018–2019) (**b**), and adult bee cluster size of the colonies was monitored in the fall and the following spring. Bars indicate mean ± SD interframe spaces occupied by the adult bee cluster for twenty to twenty-two colonies per group. Different letters indicate significant differences at each time point, *p* < 0.05 by ANOVA or Kruskal-Wallis rank test.

**Figure 4 insects-11-00030-f004:**
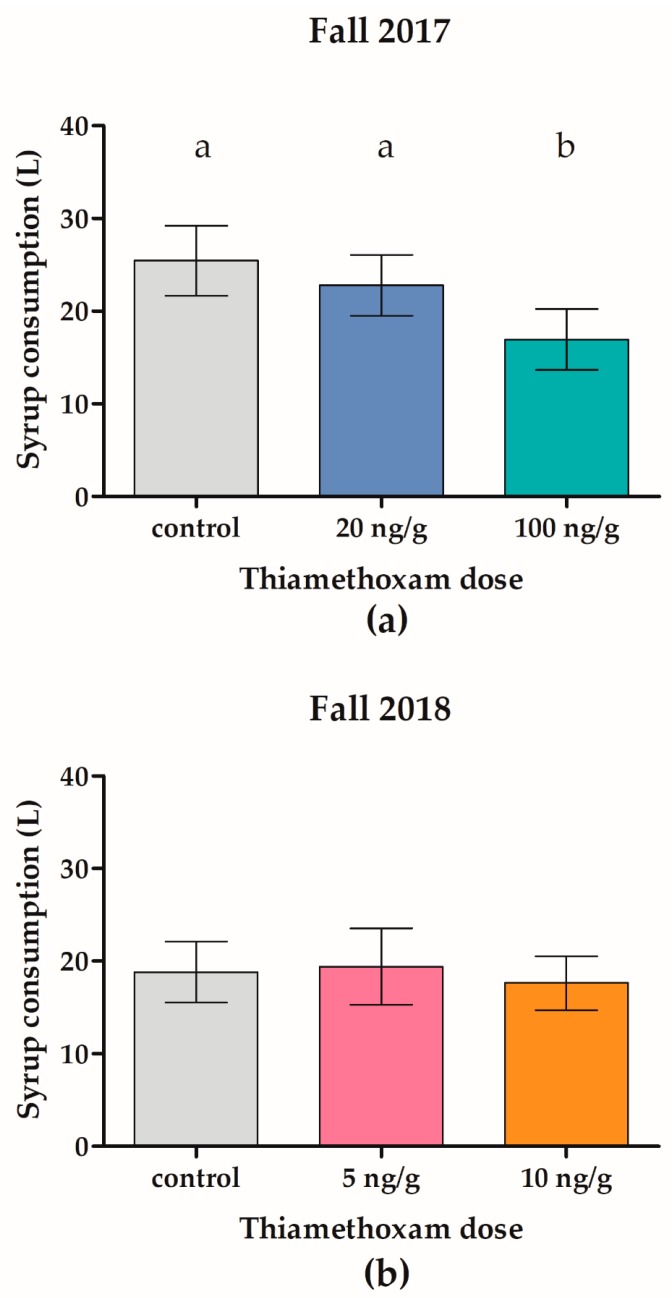
Colonies exposed to chronic high-dose (100 ng/g) thiamethoxam consumed significantly less syrup. Honey bee colonies were fed control or thiamethoxam (THI)-contaminated sugar syrup ad libitum over five weeks during fall 2017 in field trial 1 (**a**) or during fall 2018 in field trial 2 (**b**). Bars indicate mean ± SD liters of sugar syrup consumed for twenty to twenty-two colonies. Different letters indicate a significant difference, *p* < 0.001 by ANOVA.

**Figure 5 insects-11-00030-f005:**
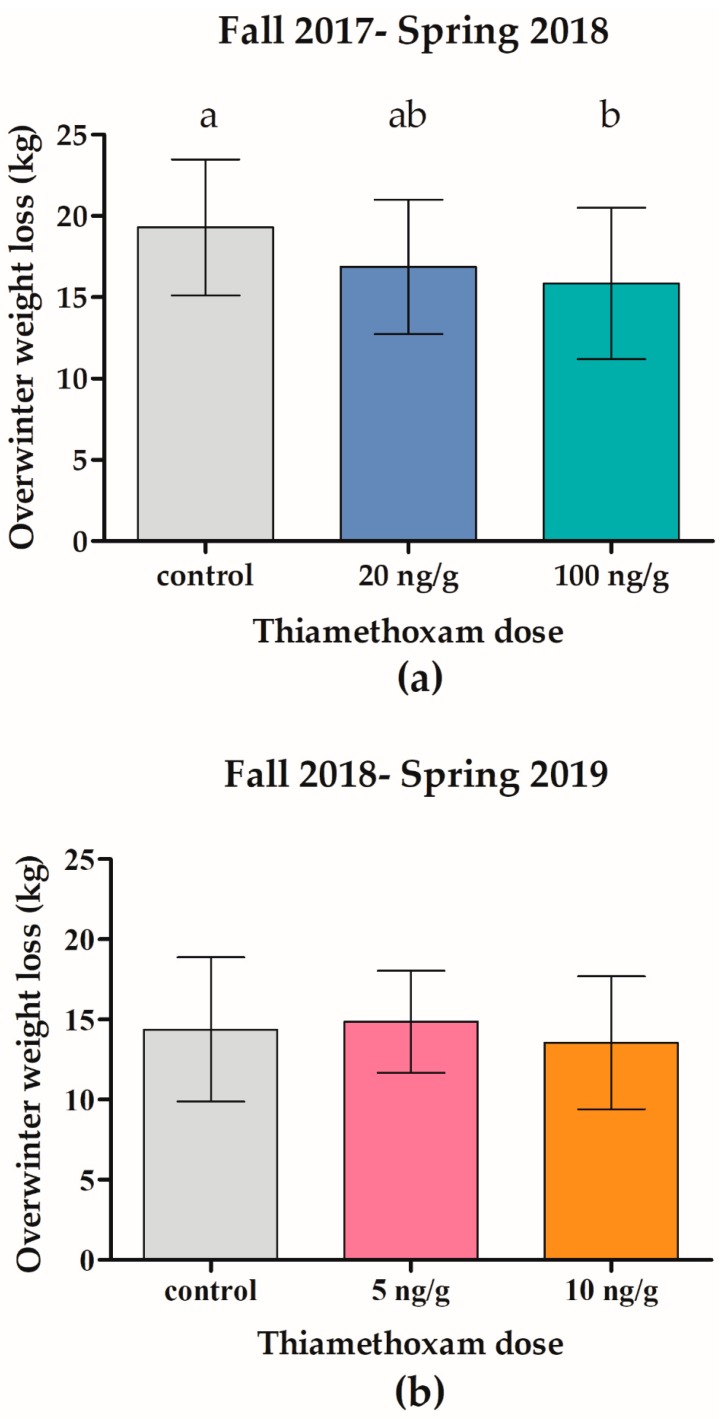
Colonies exposed to chronic high-dose (100 ng/g) thiamethoxam lost significantly less weight during winter compared to controls. Honey bee colonies were exposed to thiamethoxam (THI) during winter 2017–2018 in field trial 1 (**a**) or winter 2018–2019 in field trial 2 (**b**). Bars indicate mean ± SD overwinter (October to May) weight change in kg for twenty to twenty-two colonies. Different letters indicate significant differences, *p* < 0.05 by ANOVA.

**Figure 6 insects-11-00030-f006:**
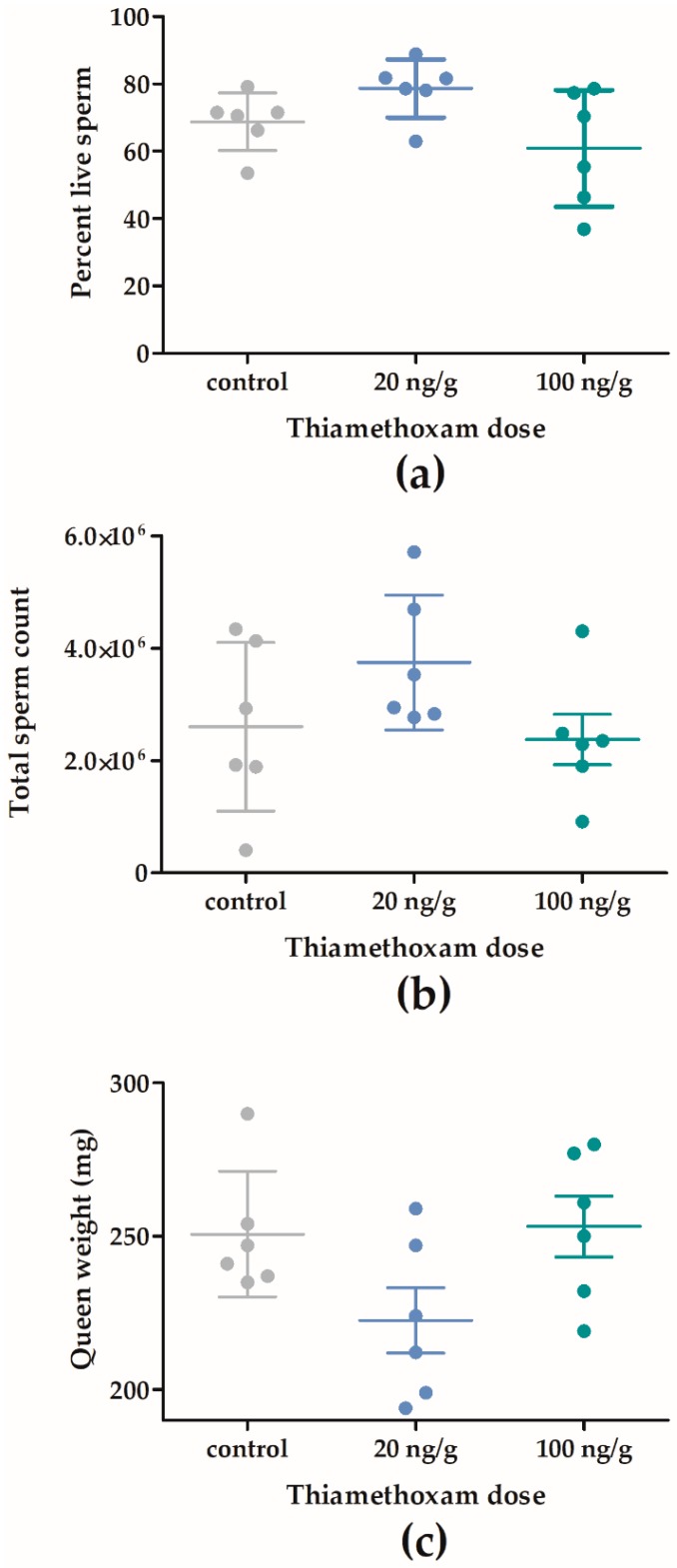
Chronic overwinter thiamethoxam (20 and 100 ng/g) exposure does not impact *Apis mellifera* queen quality. Honey bee colonies were exposed to thiamethoxam (THI) during winter 2017–2018 in field trial 1, and queens were sacrificed in August 2018, for determination of percent sperm viability (**a**), total sperm counts (**b**), and queen weight (**c**). Plots indicate mean ± SD.

**Figure 7 insects-11-00030-f007:**
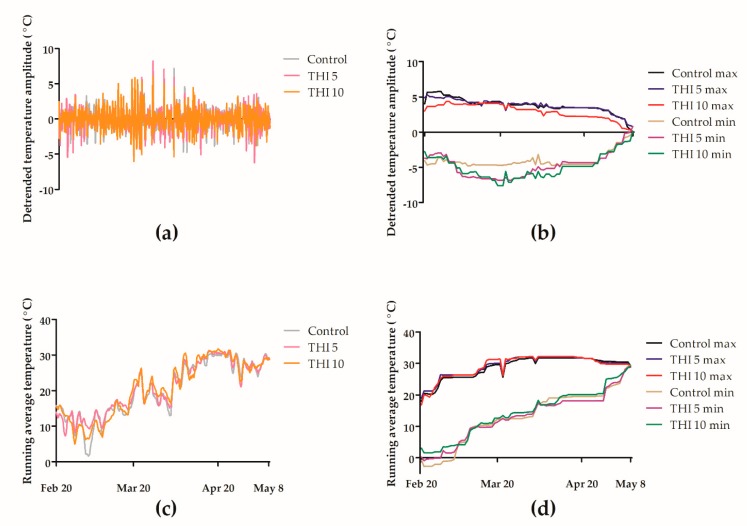
Environmental doses of (5 and 10 ng/g) thiamethoxam do not significantly affect within hive temperature during winter and early spring compared to controls. Honey bee colonies were exposed to thiamethoxam (THI) during winter 2018–2019 in field trial 2. Lines indicate mean detrended temperature amplitude (**a**), maximum and minimum detrended temperature amplitude (**b**), mean running average temperature (**c**), and maximum and minimum running average temperature (**d**), in degrees Celsius, for 10–14 colonies for 78 days, from February to May 2019.

**Figure 8 insects-11-00030-f008:**
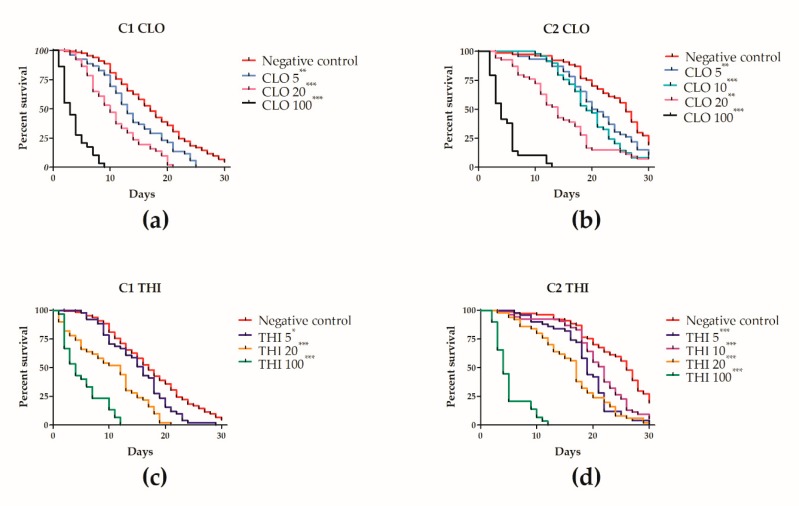
Chronic laboratory thiamethoxam or clothianidin exposure significantly decreases winter adult worker *Apis mellifera* survival. Winter workers were exposed to thiamethoxam (THI) or clothianidin (CLO) (doses in ng/g) for 30 days, through syrup, and mortality was monitored daily. Two laboratory cage trials were conducted: C1 in March–April (**a**,**c**) and C2 in April–May (**b**,**d**). Only C2 included 10 ng/g dose groups. Lines indicate percent daily survival for 131 (**a**,**c**) to 77 (**b***,***d**) bees in the negative control groups; 29–30 bees in the 100 ng/g groups; and 46–54 bees in the other treatment groups. *, **, *** survival significantly different from control, *p* < 0.05, 0.01, 0.001, by a Weibull accelerated failure time model.

**Figure 9 insects-11-00030-f009:**
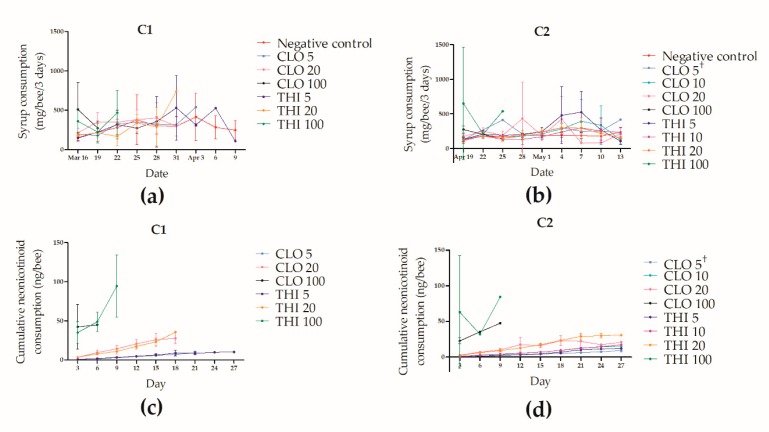
Laboratory syrup consumption and cumulative neonicotinoid consumption of winter adult *Apis mellifera* workers. Winter workers were exposed to thiamethoxam (THI) or clothianidin (CLO) (doses in ng/g) for 30 days, through syrup, and mortality was monitored daily. Two laboratory cage trials were conducted: C1 in March–April, 2018 (**a**,**c**) and C2 in April–May, 2018 (**b**,**d**). Only C2 included 10 ng/g dose groups. Lines indicate mean syrup consumption in mg per bee per three days (**a**,**b**) and mean cumulative neonicotinoid consumption in ng per bee (**c**,**d**). ^†^ Consumption data for day six were missing for the CLO 5 group in C2.

**Table 1 insects-11-00030-t001:** Experimental design and sample size (N) of field trials examining survival of outdoor overwintered colonies chronically exposed to thiamethoxam (THI).

	Field Trial 1 (F1)	Field Trial 2 (F2)
Trial dates (September–May)	2017–2018	2018–2019
THI doses (ng/g)	0, 20, 100	0, 5, 10
N (colonies) per dose	20, 20, 20	22, 21, 21
N for queen quality analysis	6, 6, 6	-
N for temperature analysis	-	14, 11, 10
*Varroa* and *Nosema* monitoring	Yes	No

**Table 2 insects-11-00030-t002:** Experimental design of laboratory cage trials examining survival of winter honey bee workers chronically exposed to thiamethoxam (THI) or clothianidin (CLO) for 30 days.

	Cage Trial 1 (C1)		Cage Trial 2 (C2)	
Trial Dates	13 March–11 April 2018		16 April–15 May 2018	
Negative control	1:1 (*w/v*) sucrose solution		1:1 (*w/v*) sucrose solution	
# negative control cages	13		8	
Positive control	1000 ng/g dimethoate		1000 ng/g dimethoate	
# positive control cages	3		3	
Mean bees per cage (SD)	10.1 (0.97)		9.9 (1.1)	
# diet evaporation cages	3		3	
Neonicotinoids tested	THI	CLO	THI	CLO
Doses (ng/g)	4.9, 19.5, 97.3	4.2, 16.7, 83.2	4.9, 9.7, 19.5, 97.3	4.2, 8.3, 16.7, 83.2
# cages per dose	5, 5, 3	5, 5, 3	5, 5, 5, 3	5, 5, 5, 3

# refers to number of cages.

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
