# Peer review of "Chronic High-Dose Neonicotinoid Exposure Decreases Overwinter Survival of Apis mellifera L."

_insects, 2019, doi:10.3390/insects11010030_

Round 1

Reviewer 1 Report

COMMENTS TO THE AUTHOR:

Summary

This manuscript examine the sublethal effects of the neonicotinoid insecticides clothianidin and thiamethoxam in field trials and under laboratory conditions. What I found interesting was 1) the combination of field study and laboratory setting, 2) the focus on the effect on overwinter survival in the relative harsh winter conditions of Sasketchewan, Canada and 3) the different assays that were applied for this study. The results suggests that high doses of insecticides do have a clear effect on the overwintering survival under field realistic conditions, but not significant change was recorded for the low “sublethal” doses. In contrast, all sublethal doses had an impact on survival for the laboratory setting. To my knowledge, this is one of the first studies dealing with sublethal effects of neonicotioids on overwintering survival of Apis mellifera in this geogrqphic setting and thus is principally worthy to be published in “Insects”.

However, the manuscript needs a lot of work with regards of the organization and clarity and should therefore be reformatted to enhance clarity. Several information with the same content are mentioned in the manuscript for several times, for example the different doses of neonicotionids used and the different field trials F1 and F2. I would strongly suggest reorganizing the Material and Methods section, splitting it into two main parts, the field and the laboratory field trials. The separation into the different sections preparation, interventions and analysis of data is not necessary, several information included here seem not to be crucial and could be removed. In addition, separate parts dealing with Varrora, Nosema and queen fitness could be added.

In the abstract part, you put too much detailed information that you will provide in the results section anyway, but you are missing other results, i.e. queen performance and temperature measurements. I would recommend to explain in more detail the hypothesis of the potential temperature difference in the colonies treated with the insecticides. If I assume correctly, the idea would be that sublethal doses of neonicotinoids will interfere somehow with the ability of the overwintering bees to maintain ambient temperatures for the colonies, that you discuss in the discussion part and which I find quite interesting.

Abstract

Line 17: It is not necessary to put all the details about the different concentrations into the abstract (0, 5, 10, 20, or 100 ng/g). Remove this part and add …with different concentrations instead.

Line 17: Here you use twenty as word, in other parts you write it as number (Line 19), please stay consistent throughout the manuscript.

Line 17-21: Instead of putting the results section here in the abstract describe in your words the effects in a more general fashion, all the information in brackets can be removed from this section.

Line 21-25: Same comment as above, get rid of the result like style and put the information in a more general context.

Generally, the focus of the abstract was just on the survival, you could add also the different other results shortly.

Introduction

Line 33: Citation for this statement is missing.

Line 39: Replace in our area with Canada.

Line 42: Remove our geographic area in

Line 44-53: This part is difficult to understand. The first sentence of the paragraph is summarizing the result, and explaining it subsequentally for each does not really enhance clarity. Please rephrase this part, with pointing out the differences of the studies you cite.

Line 58-59: Remove hemolymph in both cases and add “in the hemolymph” after vitellogenin and other proteins.

Line 73: Why is 5-20 ng/g field realistic? Before you write about up to 9.4 ng/g found Clothianidin? Reference?

Line 75-79: This section belongs to the result part and can be removed here. Still, several other hypothesis that you want to test are not listed here (termperature, queen fitness, feeding…)

Material and Methods

General question: What is your rationale to examine different concentrations on two different time points?

Line 83-84: Remove the part in brackets.

Line 92: Were the colonies at the beginning in similar condition? Equally strong? Same age of the queens?

Line 95 & 99: Are the F1 and F2 queens genetically related to each other?

Line 97 vs 101: The location of F1 and F2 based on the coordinates are approx. 180 miles apart from each other, what is the reason for this? Are the climatic conditions the same for both locations?

Line 112-113/ Table 1: It is confusing, that in the text feeding with THI was conducted for five weeks, but in table 1 the timeframe of eight month is written. The study design has to be written more clearly in this regard. Some studies were just performed for F1 or vice versa for F2, like queen quality analysis, temperature analysis and Varroa and Nosema monitoring, why is this the case? I am not sure that this is a stable base for comparisons / conclusions in general.

Line 124/ Table 2: Formate upper part of table, line missing for April 16 – May 15, 2018. In table 2, you did not mention why Dimethoate is a positive control, and also the dose should be adapted to ng/g as has been done previously for the other samples. Why did you use the mean of bees and not the total number? The doses in nM has no additional information and can be removed.

Line 126-134: This part can be deleted.

Line 136-137: Fall feeding with sugar syrup is not just a standard beekeeping practice in Saskatchewan, but everywhere were you remove the honey from the colonies. Remove the sentence.

Line 139-145: The information is already provided in table 1, shorten or remove.

Line 144: Units are not harmonized. Change to ng/g.

Line 171-172: please specify the quality oft he photo, always same settings used? Same magnification? Same distance? Same light condtions? Could you provide one photo as example? Or a scheme how you measured the cluster size?

Line 176: To compare all different conditions for cluster size, the conditions have to be the same. It cannot be that you change settings by visual assessment and compare it with the other conditions. Remove the time point 1 in F1 or describe, why it was not possible to do the cluster analysis.

Line 197: What is Kiev buffer? Please describe briefly in this section.

Line 227: In Williams et al. 2013 an optimal ratio of cage to bee is 3:1 [Williams, G.R.; Alaux, C.; Costa, C.; Csáki, T.; Doublet, V.; Eisenhardt, D.; Fries, I.; Kuhn, R.; McMahon, D.; Mderzycki, P.; et al. Standard methods for maintaining adult Apis mellifera in cages under in vitro laboratory conditions. J. Apic. Res. 2013, 52, 1–36.], so the cages you used are too big fort he amount of bees used. Probably you could discuss this point in the discussion section.

Line 245: Are those technical or biological replicates? What was the incubation temperature? What does the warm temperature do with winterbees, that should be adapted to colder temperatures?

Results

Line 308/Figure 3: First sentence in bold, all other figure legends are non-bold, please correct.

Line 364-373: The survivals were conducted in replicates, are those technical or biological replicates?

Line 383: „Table 1“ has to be „Table 3“, also in the main text body, line is just under Median survival time, is that on purpose? So the information value between Figure 7 and this table is very similar. I would suggest to put the table in the supplementary part.

Line 391: 1 mg/kg different unit again. Please correct.

Discussion

Line 418-419: THI on 20 ng/g did not significantly decrease overwinter surival, period!!!! No speculation on non-significant results!

Line 425: See comment line 418-419.

You should cite following manuscripts for the discussion part:

Genersch et al. (2010) DOI: 10.1051/apido/2010014. This publications deals with the general causes of winter lethality in honeybees with one oft he biggest factor beeing the Varroa mite.

Jacques et al. (2017) DOI: 10.1371/journal.pone.0172591. Survival of the colonies depend heavily on the training of the beekeeper.

Bartling et al. (2019) DOI:10.3390/insects10100340. Sublethal doses of neonicotinoid Clothianidin is interfering with conditioning and biosensory abilities of honeybees.

Cutler et al. 2014; DOI: 10.7717/peerj.652   Residues of Clothianidin in pollen

Reviewer 2 Report

The authors have presented a well-documented study on the impacts of two neonicotinoids – thiamethoxam and clothianidin – on colony overwintering and diutinus bee survival. The study is important and reiterates the needs to investigate field realistic impacts on overwintering honey bee colonies. However, there are significant flaws in the study design and result interpretations, for which the results and discussion sections hold no meaning. The effort of the authors are commendable as they have been working with a large number of hives. However, the flawed methods and study design have rendered the observations incorrect.

Why did the authors use different doses for different lab and field studies? The authors must use all doses during the same field trial and lab cage study, otherwise testing different doses at different times, considering variations in colony genetics (different sister queens for different years) and seasonal physiology, it is highly improbable to understand the effects. The same is true for the lab cage study. Why did the authors not follow the standard lab cage dose-response study practices? The authors must re-write the interpretations for individual field and lab cage studies. Two different field studies with two different batches of sister queens, and two different lab cage studies at two different times of the year, cannot be considered the same and thus the results cannot be merged. This essentially means if the authors have used three different concentrations for F1 and C1 (for example), and analyzed the results separately (as they should), it only tells us about those colony responses for those two doses in year 2017 only. In the results and discussion sections, responses to all doses have been described together, which is incorrect. Similarly, for lab cage studies, if the authors tested only three doses of CLO or THI for C1 or C2, they are only able to interpret that, as bees from the second study may have responded differently if all doses were tested during the same experiment. For F1 and F2, were the hives normalized at the beginning of each field study?

This also brings me to another major concern – why were unequal number of hives/cages tested? I completely understand field studies do not go as planned and the authors’ efforts are commendable in maintaining the large number of colonies for both field studies. However, the authors should have equally split the hives for all doses for both F1 and F2. The same is true C1 and C2. For cage studies, how did the authors ensure that the sampled bees were all of the same age? Did the authors paint-mark the overwintering bees to know their age cohorts? L120 29-131 number of honey bees is a huge difference. Their trophallaxis, clustering and ages play a vast role in their responses to the neonics, especially because it was administered through sugar syrups. Even the number of negative control cages varied.

What was the interval of sugar syrup supplementations for F1 and F2 studies? Why was queen health not observed in F2? Why was F1 temperature not logged? No lab cage study in 2019?

The method sections were very hard to follow. Too many variances in the study design, too many numbers and too many unequal replicates and doses. Please provide a flowchart for the benefit of the readers. I had to go back and forth a number of times to understand the study design. How did the authors maintain the lab cages? For multiple comparisons, please use alphabets to indicate significant differences in the graphs. What do the authors mean by interframe spaces?

Round 2

Reviewer 2 Report

I appreciate the authors' efforts to significantly revise their manuscript. I understand why it is not possible to revise the study design now. But for future studies, my suggestion would be to use the same number of replicates and same number of bees in each replicate, despite the dose. High mortality for higher toxic doses can be taken into account by having a large number of bees in each replicate cage (100-200). 10 honey bees per cage is too small a number for any experiment. Minor suggestions for the figures, please write the alphabets close to the bar. They do not have to be at the same height, as long as they are close to the bar and easily interpreted. Also, in future, maintain lab cages at 32 degrees C and 55%RH. These values are closer to the ideal rearing conditions of honey bees in the incubator. Frames of bees is a better method to count for adults, instead of interframe spaces. Or alternately, use imaging to estimate cluster sizes or grids for the same reasons.